# RobustGEC: Robust Grammatical Error Correction Against Subtle Context Perturbation

Yue Zhang[☆*]   Leyang Cui[ᏻ†]   Enbo Zhao[ᏻ]   Wei Bi[ᏻ]   Shuming Shi[ᏻ]

[☆]Institute of Artificial Intelligence, School of Computer Science and Technology,
Soochow University, Suzhou, China [ᏻ]Tencent AI Lab, Shenzhen, China

yzhang21@stu.suda.edu.cn

{leyangcui,enbozhao,victoriabi,shumingshi}@tencent.com

 https://github.com/hillzhang1999/RobustGEC

## Abstract

Grammatical Error Correction (GEC) systems play a vital role in assisting people with their daily writing tasks. However, users may sometimes come across a GEC system that initially performs well but fails to correct errors when the inputs are slightly modified. To ensure an ideal user experience, a reliable GEC system should have the ability to provide consistent and accurate suggestions when encountering irrelevant context perturbations, which we refer to as *context robustness*. In this paper, we introduce RobustGEC, a benchmark designed to evaluate the context robustness of GEC systems. RobustGEC comprises 5,000 GEC cases, each with one original error-correct sentence pair and five variants carefully devised by human annotators. Utilizing RobustGEC, we reveal that state-of-the-art GEC systems still lack sufficient robustness against context perturbations. In addition, we propose a simple yet effective method for remitting this issue.

## 1 Introduction

Grammatical Error Correction (GEC) is the task of automatically fixing various textual errors in a given sentence. This task has many real-life applications, such as writing assistance and language teaching, thus receiving considerable interest from both academia and industry (Grundkiewicz et al., 2020; Wang et al., 2021; Bryant et al., 2022; Zhang et al., 2023a).

Nowadays, cutting-edge GEC systems perform decently in academic benchmarks (Ng et al., 2013, 2014; Napoles et al., 2017; Bryant et al., 2019). However, in practical applications, the GEC system may offer different error correction suggestions when users perform irrelevant modifications. Figure 1 presents a real example of this phenomenon, where a user interacts with a GEC system based on T5 (Rothe et al., 2021). Given the original input "I

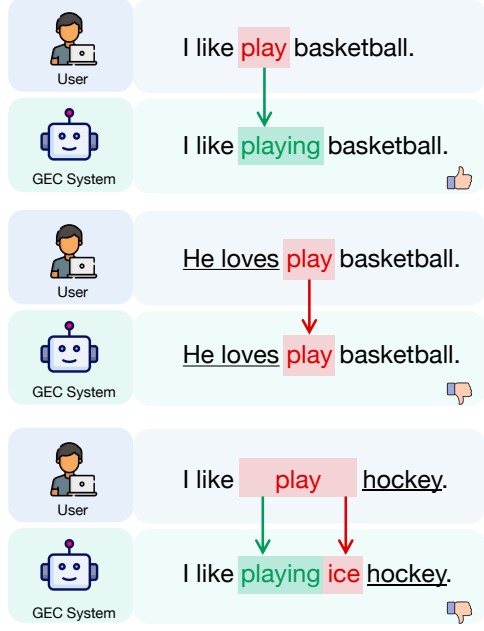

Figure 1: A real example of how minor changes in user input misleads a T5-based GEC system to wrong predictions. We use Red and Green to highlight errors and corrections, and underline to mark user modifications.

like play basketball", the GEC system effectively corrects the error (*play ⇒ playing*), because "like" should be followed by a gerund. However, when the user introduces minor modifications unrelated to the error correction, the GEC system starts making mistakes, including under-correction (*play ⇒ play*) and over-correction (*hockey ⇒ ice hockey*). This observation may lead to confusion and frustration, as users would expect the GEC system to have a more stable and reliable performance.

A key desirable property of strong GEC systems is *context robustness*, which refers to the ability to maintain consistent and accurate results without being disrupted by irrelevant context perturbations. To test this, we contribute the **RobustGEC** benchmark based on traditional GEC datasets (§3). We choose 5,000 error-correct sentence pairs from three sources for annotation, i.e., CoNLL-14 (Ng

---

*Work was done during the internship at Tencent AI Lab.

†Corresponding author.

et al., 2014), BEA-19 (Bryant et al., 2019), and our newly released TEM-8. The first two are collected from essays and corrected by native English-speaking teachers. The last is derived from the ungrammatical sentence modification question in TEM-8, a high-level exam in China for evaluating English major students' English ability, where errors are designed by experts. We ask annotators proficient in English to carefully craft five variants for each original sample by perturbing the context unrelated to the correction. The annotators are allowed to freely replace, insert, or delete content without introducing new errors or altering the original errors. In this way, RobustGEC can examine whether GEC systems consistently and accurately rectify errors disregarding *natural* and *realistic* irrelevant context perturbations.

Utilizing RobustGEC, we evaluate the context robustness of five state-of-the-art (SOTA) GEC systems (§4), i.e., *seq2seq*-based BART (Katsumata and Komachi, 2020), *seq2seq*-based SynGEC with linguistic knowledge (Zhang et al., 2022b), *seq2edit*-based GECToR (Omelianchuk et al., 2020), *LLM*-based LLaMA with GEC fine-tuning (Touvron et al., 2023), and *LLM*-based ChatGPT with zero-shot prompting (OpenAI, 2023). Despite their notable performance on conventional GEC benchmarks, all evaluated systems show dramatic performance fluctuations when facing context perturbations, revealing their lack of context robustness. Taking BART as an example, only 43.5% of GEC instances exhibit consistent corrections during perturbation. One potential explanation is that current GEC systems do not truly comprehend grammar, but rather depend on some spurious correlations (Tu et al., 2020; Yang et al., 2022) to make corrections. In order to gain more in-depth insights, we also conduct detailed analyses to investigate how context perturbations affect GEC systems, in terms of perturbing action, distance, word frequency, etc.

To improve the context robustness, we further propose a **C**ontext **P**erturbation **R**obust (CPR) post-training approach (§5). The proposed CPR method optimizes the GEC system to output the same prediction distribution at the non-perturb positions by minimizing the bi-directional Kullback-Leibler (KL) divergence, which is lightweight and easy to train. Experiments demonstrate that the CPR method can improve context robustness by a large margin and maintain the original GEC performance.

For example, after post-training with CPR, the GECToR model performs consistent corrections for 15.1% more GEC cases in RobustGEC.

We hope this research could spur future investigations into the robustness of GEC systems. To facilitate further studies, we have made all data and code publicly accessible at `https://github.com/hillzhang1999/RobustGEC`. In the context of the LLM era, we believe that there is still much more to explore for GEC, extending beyond the mere pursuit of superior leaderboard scores.

## 2 Related Work

**GEC Benchmarks.** To better study GEC, researchers have dedicated a significant effort to build solid evaluation benchmarks (Ng et al., 2014; Napoles et al., 2017; Bryant et al., 2019; Napoles et al., 2019; Flachs et al., 2020; Zhang et al., 2022a, 2023d). These benchmarks typically evaluate GEC performance using reference-based metrics like edit-level $F_{0.5}$ (Dahlmeier and Ng, 2012; Bryant et al., 2017). Despite their contributions, these benchmarks are limited in their ability to assess the robustness of GEC systems to context perturbations, as each error in them is associated with only one specific context. Therefore, we decide to build RobustGEC to fill this gap.

**GEC Approaches.** For building advanced GEC systems, there are three mainstream paradigms today. The first is sequence-to-sequence (seq2seq), which treats GEC as a monolingual machine translation task and employs encoder-decoder models such as Transformer (Vaswani et al., 2017) to generate corrections (Junczys-Dowmunt et al., 2018; Stahlberg and Kumar, 2021; Rothe et al., 2021; Zhang et al., 2022b). The second is sequence-to-edit (seq2edit), which predicts a sequence of edits and applies them to source tokens to perform corrections (Awasthi et al., 2019; Omelianchuk et al., 2020; Stahlberg and Kumar, 2020). The last is based on the burgeoning large language models (LLMs), such as ChatGPT and LLaMA (Touvron et al., 2023), which can achieve promising GEC performance via prompting (Coyne and Sakaguchi, 2023; Fang et al., 2023) or fine-tuning (Zhang et al., 2023b). In this work, we select five representative systems that cover all three paradigms for evaluating context robustness on RobustGEC.

**Robustness in Other NLP fields.** Robustness in NLP has long been an active research topic (Wang

| Source | CoNLL-14 | BEA-19 | TEM-8 |
|---|---|---|---|
| Number of Sentences | 1,312 | 2,503 | 1,185 |
| Number of Wrong Sentences | 1,176 | 1,687 | 1,184 |
| Average Length (Word) | 22.98 | 20.97 | 30.65 |
| Average Number of Errors | 2.60 | 3.12 | 1.47 |

Table 1: Statistics of our three data sources.

| ID | Case | Wrong Reason |
|---|---|---|
| 1 | I have a [lot→lots] of friend. | *Correctness* |
| 2 | He will give a talk ~~yesterday~~. | *Faithfulness* |

Table 2: Examples of bad cases generated by automatic rules. Blue highlights the automatic perturbation and Red marks the grammatical error.

et al., 2022). Using adversarial text generation (Ebrahimi et al., 2018; Ren et al., 2019; Li et al., 2020), existing work automatically builds synthetic robustness benchmarks for various tasks, such as question answering (Jia and Liang, 2017), name entity recognition (Lin et al., 2021), and sentiment analysis (Xing et al., 2020). Meanwhile, there are also a lot of studies on improving NLP models' robustness from both continuous (Chen et al., 2020; Zhu et al., 2020) and discrete (Kaushik et al., 2019; Liu et al., 2021) spaces, which may offer useful insights for building robust GEC systems.

**Robustness in GEC.** Compared with other NLP areas, research on robustness in GEC is relatively scarce. Wang and Zheng (2020) study the robustness of GEC models under the scenario of varying numbers of errors. Raina et al. (2022) instead focus on security and reveal that GEC systems can be easily fooled by appending an adversarial phrase. Unlike them, our work explores the robustness of GEC systems when nuanced modifications irrelevant to errors are introduced by users, which is most like the recent work from Feng et al. (2023). Addressing this issue is crucial, as the robustness against irrelevant variations could significantly affect users' trust in the GEC system's reliability.

## 3 RobustGEC Benchmark

### 3.1 Data Collection

To ensure a systematic and comprehensive evaluation for context robustness, we collect realistic GEC samples from three sources for annotation, i.e., CoNLL-14, BEA-19 and TEM-8.

CoNLL-14 (Ng et al., 2014) and BEA-19 (Bryant et al., 2017) are two widely-used GEC evaluation datasets. Sentences in CoNLL-14 are collected from student essays written by English-as-Secondary-Language learners, while BEA-19 comprises essays written by both learners and natives. Sentences in them are corrected by native teachers. We utilize the whole CoNLL-14 test set and a part of the BEA-19 dev set for annotation.

TEM-8 is a newly released dataset in this work.

We collect it from the "LANGUAGE USAGE" section in TEM-8 [1], the highest-level test for senior students majoring in English Language and Literature in China. TEM-8 contains multifarious challenging grammatical errors designed by linguistic experts, which aim at examining whether students meet the English language proficiency specified in the "National College English Teaching Syllabus" for English Majors. We gather about 1,000 GEC samples from this source for annotation.

The data statistics of each source are presented in Table 1. The total number of original GEC samples for annotation is 5,000. We can see that there exist some discrepancies between the three data sources regarding the average sentence length and the average number of errors, which may help RobustGEC support a more comprehensive evaluation.

### 3.2 Annotation Procedure

As mentioned in §2, one common practice in many NLP fields, e.g., text classification (Zhang et al., 2019) and named entity recognition (Lin et al., 2021), is automatically constructing synthetic benchmarks to probe model robustness. Compared with these tasks, GEC is more sensitive to additional grammatical errors introduced by synthetic context perturbations. We present two illustrative cases in Table 2 to explain why this method is not well-suited for GEC. We perturb *Case#1* with automatic synonym substitution (Zhou et al., 2021) and find this introduces a new error (*a lots of*). This occurs because "lot" and "lots" are considered synonymous, but "lots" is inappropriate in this context. We perturb *Case#2* by randomly deleting a word. However, this changes the original error since the deleted "yesterday" is the evidence of the error (*will give⇒gave*).

From these examples, it becomes evident that synthetic context perturbation often introduces new grammatical errors or alters existing ones, leading to an unreliable evaluation of robustness in GEC. To make our benchmark convincing, we ultimately

---

[1] http://tem.fltonline.cn/

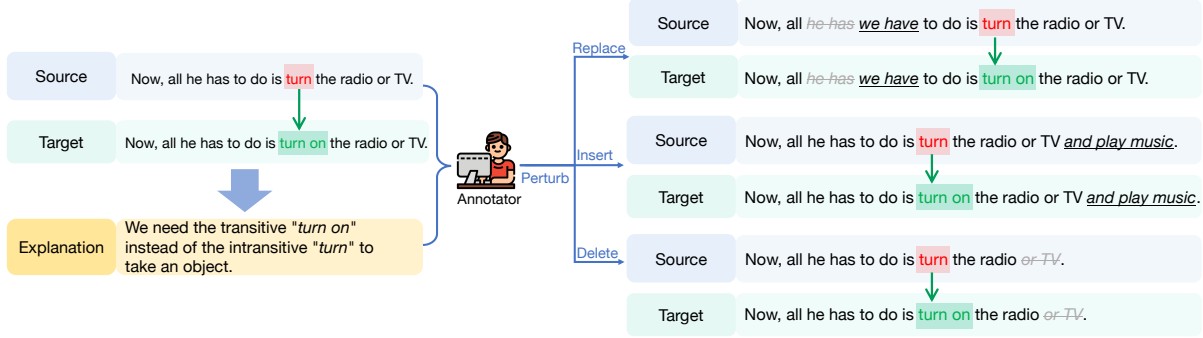

Figure 2: An illustrative example of the annotation procedure of RobustGEC.

opt for manual annotation in the creation of Robust-GEC. We engaged 10 annotators, all of whom hold either bachelor's or master's degrees in English and have successfully passed the TEM-8. The demographic features (e.g., gender, age, and homeplace) are uniformly distributed among our annotators.

Our annotation workflow is illustrated in Figure 2. During the annotation process, we employ the ERRANT toolkit (Bryant et al., 2017)[2] to align the source and target sentences of the original GEC sample, extracting errors and their corresponding corrections. We then highlight these errors using prominent colors and assign the sample to annotators for perturbation. The annotator is allowed to perturb the non-error context by replacing/deleting/inserting content freely. The perturbation should be subtle, ensuring that it neither alters the intended meaning of the sentence nor involves excessive content changes. More importantly, the perturbation must adhere to two principles:

- **Correctness**: the perturbation must NOT introduce any new errors.

- **Faithfulness**: the perturbation must NOT change the original errors.

The underlying motivation for these two principles is to emulate a situation where users make minor, error-irrelevant modifications while expecting the GEC system to maintain accurate, consistent corrections. We instruct annotators to carefully review their annotations to guarantee adherence to these principles. Each sample undergoes annotation with one perturbation by five distinct annotators. Finally, we obtain six variations of a GEC case by combining the original sample with the five perturbed versions.

[2]https://github.com/chrisjbryant/errant

**Annotation Analysis.** After annotation, there is an average of 1.29 perturbing edits per sample. This indicates that the perturbations in RobustGEC are quite unobtrusive. Nevertheless, in the later §4.2, we demonstrate that all evaluated GEC systems, despite performing well on traditional benchmarks, still remain susceptible to such subtle perturbations. In addition, we observe that annotators exhibit a preference for substitutions over insertions and deletions, and they tend to perturb nouns and verbs more frequently than other part-of-speech types, as nouns and verbs typically serve as the core components of sentences.

**Quality Assurance.** To ensure the quality of RobustGEC, we have adopted a strict quality control protocol. In the beginning, we assign a few tasks to the annotators for trial annotation. Only participants who have reached an accuracy of 90% can join the next stage. During annotation, each submission will be assigned to an experienced reviewer for double-checking. We also organize discussions regularly to address questions raised by annotators.

To further validate the annotation quality, we also conduct a human evaluation. Concretely, we ask two experienced judges to evaluate the quality of each annotation sample from *Correctness* and *Faithfulness*. We randomly select 300 annotation samples for inspection. We calculate the inter-annotator agreement (IAA) ratio of two judges and then resolve their disagreement after discussion. As shown in Table 3, most samples are acceptable considering correctness (97.3%) and faithfulness (99.3%) with high IAA ratios, demonstrating the satisfactory quality of RobustGEC.

### 3.3 Evaluation Metrics

An ideal GEC system should have both strong GEC ability and context robustness, so we conduct eval-

| | Accept (%) | Reject (%) | IAA (%) |
|---|---|---|---|
| **Correctness** | 97.3 | 2.7 | 95.6 |
| **Faithfulness** | 99.3 | 0.7 | 96.3 |

Table 3: Results of data quality inspection.

uations from these two aspects.

### 3.3.1 Metrics for GEC Ability

GEC ability refers to the performance of GEC systems in correcting errors. Following Bryant et al. (2019), we calculate precision (P), recall (R), and $F_{0.5}$ value by extracting and comparing the hypothesis and reference edits with the ERRANT toolkit (Bryant et al., 2017) to measure this ability.

Unlike previous work that calculates the similarity between a hypothesis output and one or multiple golden references, thanks to RobustGEC comprising one original sample with five corresponding perturbed variants, we can evaluate the GEC ability from multiple perspectives to gain more insights.

We report **original GEC performance** to evaluate GEC performance on the original samples, facilitating comparison with existing work evaluated on CoNLL-14 and BEA-19. Perturbations may result in either an improvement or a decline in GEC performance. Therefore, we measure **upper-bound GEC performance** and **lower-bound GEC performance** by selecting the variant that produces the highest and lowest $F_{0.5}$ values for each GEC case, respectively. The difference between upper and lower-bound performance ($\Delta F_{0.5}$) can also be considered as an indicator of context robustness.

### 3.3.2 Metrics for Context Robustness

We further devise two new metrics to measure the context robustness of GEC systems, namely **Context Robustness Score** (CRS) and **Pair-wise Context Robustness Score** (P-CRS).

CRS measures the GEC system's stability across all variants of a GEC case, indicating its ability to maintain strictly consistent corrections. CRS is formally defined as $\frac{\#\texttt{Case}_C}{\#\texttt{Case}_T}$, where the denominator is the total number of GEC cases, and the numerator is the number of GEC cases with consistent corrections for all variants.

P-CRS is more lenient and evaluates the stability between each original⇔perturb sample pair, assessing whether the GEC system can retain consistency before and after each perturbation. It can be calculated as $\frac{\#\texttt{P-sample}_C}{\#\texttt{P-sample}_T}$, where the denominator is the total number of perturbed samples, and the numer-

ator is the number of perturbed samples that are corrected consistently with the original samples.

For instance, consider a GEC case in Robust-GEC with one original sample and five perturbed samples, where four perturbed samples share the same corrections as the original sample. In such a case, CRS is $0$ while P-CRS amounts to $4/5 = 0.8$.

## 4 Evaluating Robustness of GEC Systems

In this section, we employ the RobustGEC benchmark to conduct a comprehensive test and analysis on the context robustness of existing GEC systems.

### 4.1 Selected GEC Systems

For a comprehensive evaluation, we choose five representative GEC systems that cover three mainstream paradigms as introduced in §2, including the seq2seq method—**BART** (Katsumata and Komachi, 2020), the seq2seq method with syntactic knowledge—**SynGEC** (Zhang et al., 2022b), the seq2edit method—**GECToR** (Omelianchuk et al., 2020), the LLM-based method with GEC fine-tuning—**LLaMA** (Touvron et al., 2023; Zhang et al., 2023b), and the LLM-based method in the zero-shot setting—**ChatGPT** (OpenAI, 2023; Wu et al., 2023; Fang et al., 2023).

For all systems except ChatGPT, we fine-tune them on CLang8 (Rothe et al., 2021), which is a commonly-used GEC training set with about 2.4M sentence pairs. For implementation, we adopt the authors' official code. We present more details such as hyper-parameters in Appendix A.1.

For the prompt-based method using ChatGPT, we query `gpt-3.5-turbo` (on Apr. 15, 2023) with the following prompt: "*I want you to act as a grammar checker to correct explicit mistakes in the following sentence. Please directly return the corrected sentence without explanation.*" This prompt is chosen according to our preliminary experimental results. When calling OpenAI's API, we adopt greedy search to eliminate randomness.

### 4.2 Main Evaluation Results

We present the main evaluation results for all selected systems on RobustGEC in Table 4.

**Overall Performance.** We observe that all evaluated systems, despite performing well on conventional GEC benchmarks, lack sufficient robustness against context perturbations. This is evident from their substantial performance fluctuations, as indicated by the high (>20) $\Delta F_{0.5}$ values. On the

| Model | Original | | | Upper-Bound | | | Lower-Bound | | | $\Delta F_{0.5}$ | CRS ↑ | P-CRS ↑ |
|---|---|---|---|---|---|---|---|---|---|---|---|---|
| | **P** | **R** | **F$_{0.5}$** | **P** | **R** | **F$_{0.5}$** | **P** | **R** | **F$_{0.5}$** | | | |
| *RobustGEC-Total* | | | | | | | | | | | | |
| **BART** | 55.06 | 37.92 | 50.49 | 65.38 | 42.30 | 58.95 | 35.78 | 31.44 | 34.82 | 24.13 | 43.5 | 85.4 |
| **SynGEC** | 56.21 | 38.81 | 51.59 | 66.30 | 43.17 | 59.89 | 36.14 | 31.87 | 35.20 | 24.69 | 43.2 | 85.3 |
| **GECToR** | 54.99 | 34.96 | 49.34 | 65.26 | 39.11 | 57.56 | 35.54 | 28.49 | 33.87 | 23.69 | 46.8 | 86.9 |
| **LLaMA** | 51.83 | 37.83 | 48.26 | 62.79 | 42.94 | 57.47 | 31.99 | 29.90 | 31.55 | 25.92 | 38.9 | 83.3 |
| **ChatGPT** | 36.94 | 49.50 | 38.91 | 47.29 | 56.07 | 48.82 | 23.03 | 40.49 | 25.20 | 23.62 | 20.5 | 75.2 |
| *CoNLL-14-Subset* | | | | | | | | | | | | |
| **BART** | 47.32 | 29.66 | 42.29 | 57.63 | 33.15 | 50.21 | 31.89 | 25.23 | 30.29 | 19.98 | 43.4 | 86.8 |
| **SynGEC** | 48.04 | 30.27 | 43.00 | 59.08 | 33.85 | 51.42 | 31.89 | 25.25 | 30.30 | 21.12 | 41.4 | 86.5 |
| **GECToR** | 43.83 | 30.55 | 40.33 | 52.77 | 33.98 | 47.51 | 31.39 | 26.06 | 30.16 | 17.35 | 45.4 | 87.6 |
| **LLaMA** | 43.72 | 30.38 | 40.19 | 55.29 | 34.74 | 49.44 | 27.62 | 24.16 | 26.85 | 22.59 | 35.1 | 82.9 |
| **ChatGPT** | 31.62 | 38.32 | 32.77 | 42.55 | 45.22 | 43.06 | 19.51 | 30.45 | 21.02 | 22.04 | 17.0 | 74.3 |
| *BEA-19-Subset* | | | | | | | | | | | | |
| **BART** | 62.95 | 43.68 | 57.84 | 72.78 | 48.24 | 66.06 | 40.14 | 36.60 | 39.38 | 26.68 | 41.5 | 83.4 |
| **SynGEC** | 63.87 | 44.52 | 58.76 | 72.90 | 49.00 | 66.42 | 40.30 | 36.97 | 39.59 | 26.83 | 41.8 | 83.3 |
| **GECToR** | 63.26 | 39.86 | 56.61 | 73.28 | 44.05 | 64.69 | 40.49 | 32.72 | 38.65 | 26.04 | 46.1 | 85.1 |
| **LLaMA** | 61.44 | 42.90 | 56.55 | 71.78 | 47.95 | 65.29 | 38.13 | 34.46 | 37.33 | 27.96 | 40.4 | 82.3 |
| **ChatGPT** | 40.50 | 53.94 | 42.62 | 50.90 | 60.11 | 52.51 | 25.75 | 45.35 | 28.18 | 24.33 | 21.3 | 75.2 |
| *TEM-8-Subset* | | | | | | | | | | | | |
| **BART** | 47.19 | 36.79 | 44.67 | 58.34 | 41.40 | 53.93 | 30.29 | 28.62 | 29.94 | 23.99 | 48.6 | 88.2 |
| **SynGEC** | 49.41 | 38.28 | 46.69 | 60.36 | 42.68 | 55.74 | 31.60 | 29.93 | 31.25 | 24.49 | 48.7 | 88.1 |
| **GECToR** | 47.96 | 32.53 | 43.81 | 59.56 | 36.96 | 53.07 | 29.76 | 25.61 | 28.82 | 24.25 | 50.5 | 89.9 |
| **LLaMA** | 42.63 | 37.13 | 41.40 | 53.48 | 42.75 | 50.92 | 24.89 | 27.62 | 25.39 | 25.53 | 40.4 | 85.8 |
| **ChatGPT** | 35.98 | 56.71 | 38.81 | 45.79 | 62.40 | 48.36 | 21.46 | 44.78 | 23.95 | 24.41 | 22.9 | 76.2 |

Table 4: Main evaluation results on the RobustGEC benchmark. The P/R/F$_{0.5}$ metrics are all calculated using ERRANT. $\Delta F_{0.5}$ is the absolute difference between the upper and lower bound of the system's F$_{0.5}$ score.

entire RobustGEC benchmark, all systems achieve a CRS below 50, suggesting that they would provide inconsistent corrections for more than half of our cases. Regarding the P-CRS metric, the highest value is only 86.9. These observations confirm that there is still considerable room for existing GEC systems to improve their context robustness.

**Performance on Different Systems.** As reflected by the higher CRS and P-CRS scores, GECToR consistently exhibits better context robustness compared to other systems. One possible explanation for this could be that GECToR has an encoder-only sequence labeling architecture and predicts target edits independently, whereas the seq2seq models will be affected by the context perturbations twice at both the encoder and decoder sides. Incorporating linguistic knowledge (SynGEC) does not further enhance the context robustness of BART. We speculate that this may be because the syntactic parser in SynGEC for providing linguistic information will also be affected by context perturbations. Another noticeable phenomenon is that the LLM-

based methods, including fine-tuned LLaMA and zero-shot ChatGPT, exhibit inferior context robustness compared with conventional methods. Although LLMs possess more knowledge than small-size models like BART, they also suffer from hallucinations and other issues more severely (Bang et al., 2023; Zhang et al., 2023c), which may result in instability. Given that more and more individuals rely on LLMs as their personal grammar checkers today[3], improving the context robustness of LLM-based GEC systems is an important and meaningful future challenge for GEC study.

**Performance on Different Subsets.** We also list in detail the evaluation results of all systems on different subsets. The systems collectively achieve their best context robustness on TEM-8. As mentioned before, TEM-8 is sourced from an official exam, so errors in this subset are challenging and have deterministic correction methods. We find GEC systems tend to exhibit more stable perfor-

---

[3] https://becomeawritertoday.com/chatgpt-vs-grammarly

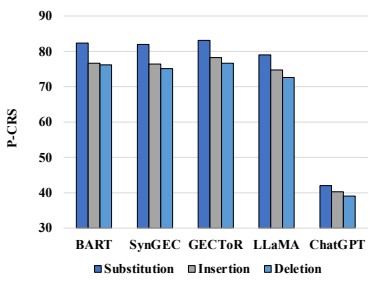

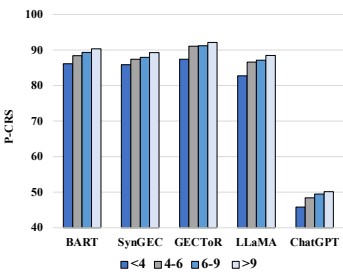

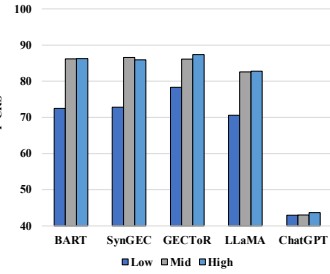

| (a) Influence of perturbing actions. | (b) Influence of perturbing positions. | (c) Influence of word frequency. |

Figure 3: Fine-grained analysis of how context perturbations affect GEC systems, as reflected by the P-CRS score.

mance for such errors compared to the expression-related errors in the other two subsets. In addition, on BEA-19, although most systems achieve their best GEC performance, their context robustness is actually the worst. This suggests that the GEC performance and context robustness may not necessarily be positively correlated.

### 4.3 Fine-grained Analysis

To gain deeper insights, we categorize context perturbations from different perspectives and employ the P-CRS score to probe whether they alter the original predictions of GEC systems.

**Influence of perturbing action.** We begin by exploring the impact of the perturbing action type. The effects of the perturbation action type are shown in Figure 3a. In comparison to substitution actions, insertion and deletion actions yield significantly lower P-CRS scores, implying that GEC systems are more likely to produce inconsistent corrections when users insert or delete irrelevant content. The insertion and deletion actions often alter the sentence structure. Such changes in sentence structure may easily mislead GEC systems into unstable predictions.

**Influence of perturbing position.** We proceed to analyze the impact of the position of the perturbation. Specifically, we only consider the perturbed sample with only one error and one perturbation. We then calculate the absolute word distance between the perturbation and the target error. Figure 3b shows the results. We observe that GEC systems are more prone to perform inconsistent corrections when the perturbation is closer to the error. Words that are close together often belong to the same sentence constituent and are closely related in terms of grammar and semantics, which could explain why perturbing them has a greater impact.

**Effect of perturbing word frequency.** We conduct an experiment to study the effect of the word frequency of the perturbations. For simplification, we only consider the substitution perturbation (e.g., replace A with B) and count the number of occurrences of the target word (i.e., B) in the training data. We categorize perturbations into three levels: *low* (less than 10 occurrences), *medium* (10 to 50 occurrences), and *high* (more than 50 occurrences). As shown in Figure 3c, all supervised systems (except the zero-shot ChatGPT) exhibit significantly low context robustness when the target word rarely appears in the training data. This phenomenon discloses that existing GEC systems may highly rely on spurious correlation patterns learned from training data rather than real grammar knowledge to fix errors. As a result, they perform inconsistently when generalizing to unseen contexts.

## 5 Improving Robustness of GEC Systems

After witnessing the unsatisfactory context robustness of existing GEC systems, we further explore the approach to improving it. In this section, we try to propose a simple yet effective method to enhance the context robustness of GEC systems.

### 5.1 Context Perturbation Robust Training

We propose a Context Perturbation Robust (CPR) training method. The basic idea is to constrain the GEC system to output the same results disregarding the perturbations in contexts. To this end, our method compares an original sample $(x, y)$ with a perturbed sample $(x', y')$, and forces the GEC model to output the same prediction distribution at the non-perturb positions by minimizing the bidirectional Kullback-Leibler (KL) divergence.

For instance, we feed the original sample $(x, y) = (x^1 x^2 x^3, y^1 y^2 y^3)$ and the perturbed sample $(x', y') = (x^{1'} x^2 x^3 x^{4'}, y^{1'} y^2 y^3 y^{4'})$ into the

| Model | O-P | O-R | O-F$_{0.5}$ | U-P | U-R | U-F$_{0.5}$ | L-P | L-R | L-F$_{0.5}$ | CRS ↑ | P-CRS ↑ |
|---|---|---|---|---|---|---|---|---|---|---|---|
| *GECToR (Baseline)* | 54.5 | 33.4 | 48.4 | 65.8 | 37.7 | 57.3 | 35.7 | 27.8 | 33.8 | 46.6 | 87.5 |
| **GECToR + CPR (ours)** | 60.9 | 27.1 | 48.6 | 70.4 | 34.1 | 58.0 | 38.8 | 22.1 | 33.7 | **61.7** (+15.1) | **90.3** (+2.8) |
| w/o KL loss | 57.5 | 29.8 | 48.3 | 72.3 | 32.3 | 58.0 | 37.4 | 22.6 | 33.4 | 50.9 (+4.3) | 87.9 (+0.4) |
| Real→Synthetic | 60.7 | 26.4 | 48.2 | 70.5 | 31.9 | 56.8 | 38.0 | 22.3 | 33.3 | 55.9 (+9.3) | 88.8 (+1.3) |
| **GECToR + Inf. Tweak** | 61.1 | 27.9 | 49.4 | 71.2 | 33.9 | 58.4 | 38.1 | 23.9 | 34.1 | 51.3 (+4.7) | 89.0 (+1.5) |

Table 5: Main results of our Context Perturbation Robust (CPR) training method for improving the robustness of the GECToR model. O/U/L-P/R/F$_{0.5}$ denote the original/upper-bound/lower-bound P/R/F$_{0.5}$ values, respectively.

GEC model, where $x^{1'}x^{4'}$ and $y^{1'}y^{4'}$ are the perturbed tokens at the source/target sides. Subsequently, we can get two prediction distributions of two samples, denoted as $\mathcal{P}(y|x) = p^1p^2p^3$ and $\mathcal{Q}(y'|x') = q^1q^2q^3q^4$. Finally, our method calculates the bi-directional KL-divergence loss at the non-perturb positions 2 and 3 as follows:

$$
\begin{aligned}
\mathcal{L}_{KL} &= \mathcal{L}_{KL}(p^2||q^2) + \mathcal{L}_{KL}(p^3||q^3) \\
&= \frac{1}{2}(\mathcal{D}_{KL}(p^2||q^2) + \mathcal{D}_{KL}(q^2||p^2)) \\
&+ \frac{1}{2}(\mathcal{D}_{KL}(p^3||q^3) + \mathcal{D}_{KL}(q^3||p^3))
\end{aligned} \quad (1)
$$

The original learning objective is the negative log-likelihood loss function, formally defined as:

$$
\mathcal{L}_{NLL} = -\log\mathcal{P}(y|x) - \log\mathcal{Q}(y'|x') \quad (2)
$$

The final training object of our CPR method is to minimize $\mathcal{L}$, which can be calculated as:

$$
\mathcal{L} = \mathcal{L}_{NLL} + \alpha \cdot \mathcal{L}_{KL} \quad (3)
$$

where $\alpha$ is the coefficient weight to control $\mathcal{L}_{KL}$.

## 5.2 Experimental Setup

For calculating the KL-divergence loss, we concatenate the original samples and the corresponding perturbed samples to form a training batch. We align them using the edit distance algorithm to achieve the perturb positions. We utilize real perturbed samples in our method for better results. Consequently, we divide RobustGEC into train/dev/test splits with 3,000/500/2,500 GEC cases, respectively. We test our method on top of GECToR. As mentioned in §4.2, GECToR has better context robustness than other systems, so further enhancing it is more challenging. We set the weight $\alpha$ of $\mathcal{L}_{KL}$ as 1.0. More details can be found in Appendix A.2.

## 5.3 Results and Analysis

**Our method improves context robustness by large margins.** We present the primary results in Table 5. As observed, post-training GECToR with our CPR method on real original⇔perturbed samples significantly enhances its context robustness. The CRS and P-CRS scores improve by +15.1 and +2.8 points, respectively. Notably, the post-training does not affect the GEC ability of GECToR, as the changes in O/U/L-F$_{0.5}$ values are minimal. These results demonstrate the effectiveness of our proposed CPR method as a lightweight plug-and-play technique for improving context robustness. We also perform a case study in Appendix C.

**Post-training without explicit constraints is insufficient.** We also report the results of removing the KL-divergence loss (w/o KL loss), which entails post-training on original⇔perturbed samples without an explicit constraint. After removing the KL-divergence loss, the boosts in CRS and P-CRS metrics become considerably smaller than before. This finding highlights the importance of explicitly enforcing the GEC system to produce consistent results using the KL-divergence loss. Furthermore, we examine the impact of the weight factor $\alpha$ of the KL-divergence loss in Appendix D.

**Synthetic perturbation data is also useful but not as effective as real data.** In previous experiments, we utilized human-annotated perturbed samples from RobustGEC. We can also generate perturbed samples using heuristic rules, which are easier to obtain. For comparison, we generate the same amount of synthetic samples by randomly substituting/inserting/deleting contents. More details can be found in Appendix B. As shown in Table 5 (Real→Synthetic), the improvement of context robustness diminishes after using synthetic data (+15.1→+9.3 for CRS, +2.8→+1.3 for P-CRS), revealing a non-negligible gap between real and synthetic perturbations. Nonetheless, the improvement remains significant, indicating that our method has

the potential to be extended to scenarios where human-annotated perturbation data is unavailable.

**The effectiveness of our method does not come from minimal modification.** As observed, our method improves the GEC model's precision but reduces recall. It is also possible that our boost in context robustness could stem from a more conservative correction trend because there exists a shortcut—if a GEC system does not correct at all, it will achieve full robustness scores as its results are very "consistent". To discuss this, we compare our method with directly tweaking the inference of GECToR, which involves adding a bias to the probability of the KEEP tag to avoid changing the source token. From the results (GECToR + Inf. Tweak), we observe that tweaking the inference leads GECToR to achieve a similar precision/recall trade-off as us. However, the improvement in context robustness is less pronounced under this setting. This indicates that the gains in context robustness we achieved are not solely due to making the GEC system correct more conservatively, but also come from teaching it to tolerate irrelevant perturbations.

## 6 Conclusion

This paper investigated the *context robustness* of GEC systems. This is the ability to provide consistent correction suggestions when modifications unrelated to errors are introduced into the inputs. To quantitatively analyze this, we developed a diagnostic benchmark named RobustGEC. Using RobustGEC, we disclosed that popular GEC systems have significant room for improvement in context robustness. Furthermore, we proposed a simple yet effective method for enhancing context robustness. We hope our work can offer insights for future research aimed at developing robust GEC systems.

## Acknowledgement

We would like to thank the anonymous reviewers and the meta reviewer for their valuable comments. We want to thank Prof. Zhenghua Li for his insightful suggestions. We also want to thank Dr. Xing Wang for helping us conduct the annotation experiment as mentioned in Appendix E. This work was partially supported by the Project Funded by the Priority Academic Program Development of Jiangsu Higher Education Institutions.

## Limitations

**Benchmark Limitation.** The scope of the data source of RobustGEC may be somewhat limited. We only consider English GEC, while there have been many studies in other languages like Chinese. Additionally, the three data sources we selected pertain only to formal writing, whereas informal writing also has a great demand for GEC. Given the aforementioned flaws, we plan to continuously update and improve RobustGEC in the future.

**Method Limitation.** Regarding our approach to enhancing context robustness, the primary limitation is that we only test it on a single type of GEC system, specifically, GECToR. In reality, our CPR method is model-agnostic, and we intend to evaluate its effectiveness on a broader range of model structures in the future. Furthermore, our method is just a simple and preliminary attempt. We hope future research will explore more effective techniques for improving robustness by using RobustGEC.

## Ethics Statement

**Data License.** When building RobGEC, we collect data from three sources, namely CoNLL-14, BEA-19, and TEM-8. The first two are publicly available GEC datasets, and the application of them in our study is consistent with their intended use and license. As for the TEM-8 data, we have consulted with professional legal advisors before collecting them. The advisors have confirmed that gathering exam questions from official institutional organizations on external educational websites does not infringe upon copyright laws, provided that no additional analytical content from these external sites is used. Besides, we do not use the outputs from test-takers, only gold answers are used. So there is no privacy problem for test-takers. We commit that all data will be used only for research purposes.

**Annotation Payment.** All annotators involved in our annotation were paid properly according to their annotated task numbers and quality. The average salary is about 60 RMB per hour, which is much higher than the legal standard in our country.

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

## A More Implementation Details

### A.1 Implementation of GEC Systems

The implementation details of the five GEC systems evaluated on RobustGEC are listed below. All experiments are conducted using 8 Nvidia Tesla-V100 32 GB GPUs. Most experiments could be complemented within several hours.

**BART.** BART (Lewis et al., 2020) is a pre-trained encoder-decoder model which is widely used for sequence-to-sequence (Seq2Seq) modeling. The pre-training task of BART is re-constructing the corrupted texts. Katsumata and Komachi (2020) has proven that BART can achieve competitive performance on conventional GEC benchmarks. We employ the `Fairseq`[4] toolkit (Ott et al., 2019) to fine-tune a BART-large[5] model (Lewis et al., 2020) on the CLang8[6] GEC training data (Rothe et al.,

---
[4] https://github.com/facebookresearch/fairseq
[5] https://huggingface.co/facebook/bart-large
[6] https://github.com/google-research-datasets/clang8

2021). This model has about 406M parameters. The training hyperparameters we used are on par with Katsumata and Komachi (2020). For decoding, we use the beam search algorithm with a beam size of 12.

**SynGEC.** SynGEC (Zhang et al., 2022b) is a GEC system built on the top of BART. It additionally incorporates tailored dependency syntax information into BART by using a GEC-oriented parser, and demonstrates such linguistic knowledge can further lead to substantial performance improvements. We reproduce the SynGEC system using their official code[7]. We also train this model on CLang8 using the default hyper-parameters as in their paper (Zhang et al., 2022b).

**GECToR.** GECToR (Omelianchuk et al., 2020) is a sequence-to-edit (Seq2Edit) GEC system that corrects ungrammatical sentences via predicting edits such as keeping and deleting. Unlike previous Seq2Seq methods, it features only a BERT-based encoder followed by a softmax layer for prediction, which makes its prediction speed very fast. We train GECToR with their official implementation[8]. We choose the Roberta-Large model (Liu et al., 2019) as the encoder of GECToR, which has about 354M parameters. Before training, we convert the error-correct sentence pairs in CLang8 to labels using the tool provided by the official GitHub repository. The hyper-parameters for training and predicting directly refer to the original paper (Omelianchuk et al., 2020).

**LLaMA.** LLaMA (Touvron et al., 2023) is an open-sourced large language model (LLM) that exhibits remarkable performance across various tasks. It is a decoder-only model with a vast number of parameters from 7B to 65B. Zhang et al. (2023b) found that LLaMA could achieve decent GEC ability after fine-tuning with GEC training data. Following Zhang et al. (2023b), we fine-tune LLaMA-7B (Touvron et al., 2023) on the CLang8 training data. Considering the time and resource costs, we conduct parameter-efficient fine-tuning by utilizing the low-rank adaptation technique (Hu et al., 2021). The hyper-parameters are directly taken from Zhang et al. (2023b).

---
[7] https://github.com/HillZhang1999/SynGEC
[8] https://github.com/grammarly/gector

**ChatGPT.** Recently, many works have shown that ChatGPT[9] can achieve promising GEC performance by using only zero/few-shot prompts (Wu et al., 2023; Fang et al., 2023; Loem et al., 2023). In this work, we also take the prompt-based GEC approach with ChatGPT into consideration. The prompt we used is described in §4.2. We set the temperature parameter as 0.0, which means we perform the greedy decoding. The reasons are two-fold. First, this will make the results fixed and reproducible. Second, we find this will lead to better GEC performance.

## A.2 Implementation of CPR Training

During the CPR post-training, we combine the original sample with each of the corresponding perturbed samples to form a contrastive pair. We implement GECToR and our CPR method with the ALLENNLP toolkit (Gardner et al., 2018)[10]. We train GECToR on CLang8 until convergence as our baseline and post-train it using our CPR method on the train split of RobustGEC. During training, we set the learning rate as 1e-6, the batch size as 128, the dropout rate as 0.3, and the weight $\alpha$ of $\mathcal{L}_{KL}$ as 1.0. The experiments are conducted on 8 Nvidia Tesla-V100 32 GB GPUs, and the results in Table 5 are averaged over 3 runs with different random seeds. The training cost is relatively inexpensive and only takes approximately 20 minutes, thereby our method is lightweight and easy to train.

## B Synthetic Perturbation Details

As described in § 5.3, we generate synthetic perturbed samples using predefined rules for comparison with real human-annotated perturbed samples. We take 3,000 original GEC samples from the train split of RobustGEC as the seed corpus and create five synthetic perturbed samples for each, ensuring a fair comparison. Each sample is perturbed only once, with substitution, insertion, or deletion applied randomly with equal probability. For substitution, we randomly select a word in the non-error context, mask it, and then use a masked language model, such as Roberta, to fill in the masked slot. For insertion, we pick a random token from the vocabulary and insert it at a random non-error position. For deletion, we randomly remove a word from the non-error context.

---

[9] https://chat.openai.com
[10] https://github.com/allenai/allennlp

| GECToR Baseline | | |
|---|---|---|
| O-S | Such people never bump up other people. | |
| O-T | Such people never bump into other people. | ✓ |
| O-H | Such people never bump into other people. | |
| P-S | Such people never bump up other people because they are very careful. | |
| P-T | Such people never bump into other people because they are very careful. | ✗ |
| P-H | Such people never bump up other people because they are very careful. | |
| **GECToR after CPR post-training** | | |
| O-S | Such people never bump up other people. | |
| O-T | Such people never bump into other people. | ✓ |
| O-H | Such people never bump into other people. | |
| P-S | Such people never bump up other people because they are very careful. | |
| P-T | Such people never bump into other people because they are very careful. | ✓ |
| P-H | Such people never bump into other people because they are very careful. | |

Table 6: A real case shows that post-training GECToR with our proposed CPR method makes it output consistent and accurate correction. O/P-S/T/H denotes Original/Perturbed-Source/Target/Hypothesis. We use Red and Green to highlight the grammatical error and the correct modification, respectively. We also use the underwave line to mark the perturbation.

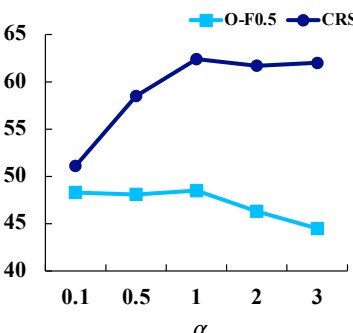

Figure 4: Original $F_{0.5}$ and CRS curves with different $\alpha$.

## C Case Study

We present a case study in Table 6. As can be seen, the GECToR baseline successfully corrects the grammatical error (up ⇒ into) in the original sample, but it fails to correct this error when an insertion perturbation is introduced. However, after post-training GECToR with our method, it can consistently and accurately correct errors in both original and perturbed samples, thereby demonstrating the effectiveness of our method.

## D Analysis of the Weight of KL Loss

We conduct an experiment to investigate the impact of the weight factor $\alpha$ of the KL-divergence loss

$\mathcal{L}_{KL}$ in Eq. 3. The results are illustrated in Figure 4. As the value of $\alpha$ increases, we observe that the context robustness of the GEC system improves, while the GEC ability deteriorates. It is also evident that setting $\alpha$ to 1.0 achieves the best trade-off. Consequently, we set $\alpha$ to 1.0 in our experiments.

## E   Comparison between Native and Non-native Annotators

It is important to highlight that the annotators involved in constructing RobustGEC are not native English speakers. To examine the potential impact of the annotators' native language on the data diversity, we launched an experiment that engaged native English speakers to perform annotations. In our preliminary annotation, which includes 100 cases, we compared the perturbations annotated by both native and non-native speakers. Interestingly, we found no significant linguistic differences, such as in edit actions and POS-tags, between the two groups of annotators. The CRS and P-CRS for BART are 41.7/83.5 on data annotated by non-native speakers and 40.8/82.8 on data annotated by native speakers, indicating only a minor difference. These findings suggest that, despite our annotators being non-native English speakers, their English proficiency is sufficiently high, enabling them to produce annotations comparable to those of native speakers.