# OpenReview forum: "RobustGEC: Robust Grammatical Error Correction Against Subtle Context Perturbation"
_EMNLP/2023/Conference — EMNLP 2023 Main_

### Official Review · Reviewer_hW9g · 2023-08-04

**Soundness:** 3

**Excitement:**

4: Strong: This paper deepens the understanding of some phenomenon or lowers the barriers to an existing research direction.

**Paper Topic And Main Contributions:**

This paper describes an ensemble data set and benchmark measuring the robustness of Grammatical Error Correction (GEC) systems against changes to the context around a grammatical error.  The authors use two pre-published GEC data sets and create a third using responses to a L2 English exam administered in China along with human annotation.  For all three data sets, humans create additional examples from those containing grammatical errors by altering the context around the error slightly.  The authors report results on this data set on five pre-published GEC systems and propose an improvement to the top-performing system.

**Questions For The Authors:**

In section 3.1 (or link to an appendix), can you go into more detail on how you sampled data from TEM-8?  Was this random or were you sampling stratified on anything such as demographics, learner-level, etc.?

When describing your inter-annotator agreement in section 3.2 under "Quality Assurance", can you specify which metric you used to measure inter-annotator agreement?

**Reasons To Accept:**

The work is thorough and the paper is well-organized.  It is pretty well-written although I have some corrections (below).  It is a great fit for the NLP Applications track.

**Reasons To Reject:**

This paper would find a very receptive audience at BEA, but unfortunately BEA 2023 already happened and the authors may not have been able to travel to Toronto.

**Reproducibility:**

4: Could mostly reproduce the results, but there may be some variation because of sample variance or minor variations in their interpretation of the protocol or method.

**Reviewer Confidence:**

5: Positive that my evaluation is correct. I read the paper very carefully and I am very familiar with related work.

**Typos Grammar Style And Presentation Improvements:**

On line 014, "cases" should be "examples".

On line 026, "application scenarios, such as writing aid" should be "applications, such as writing assistance".

On line 056, replace "i.e., CoNLL-14" with "annotation: CoNLL-14".  "i.e." is intended for examples, not for listing out the actual things.  This also applies on line 075.

On line 059, insert "English-speaking" after "native".

On line 174, "systematical" should just be "systematic".

On line 208, ", named entity recognition" should be " and named entity recognition".

On line 210, "Different from them" should be "Compared to these tasks".

On line 310, insert "the" before "original".

On line 312, "in" should be "that uses".

On line 368, "like" should be "including".

On line 400, "and predict" should be "that predicts".

On line 411, "much" is unnecessary and slightly awkward (sorry).

On line 412, "than" should be "compared to".

On lines 431-432, remove "on BEA-19," from where it is and insert it after "performance".

On line 432-433, insert "with this data set" after "robustness".

On line 453-454, "the perturbed sample" should be "perturbed samples".

On line 486-487, replace "We propose a" with "With our", insert a comma after "method", and make "The" lower-case to have this be one sentence.

On line 518, "utilize" should be "use".

On line 574, "Because there" should be "There".

On like 607, "like" should be "including".  (Using "like" in this way is too informal for this style of writing.)

On line 624, "RobGEC" should be "RobustGEC".

---

> ### Author Rebuttal · Authors · 2023-08-29
>
> ### Dear Reviewer hW9g
>
> Thank you for your hard work. Below are our responses to your concerns.
>
> >Q: This paper would find a very receptive audience at BEA, but unfortunately BEA 2023 already happened and the authors may not have been able to travel to Toronto.
>
> A:  We appreciate the reviewer's recognition that our paper would find a receptive audience at the BEA2023. Unfortunately, the deadline for BEA2023 was May 2, at which point we had not yet completed this paper. We believe that our work would appeal to NLP researchers with a focus on robustness and grammatical error correction, which is also suitable for EMNLP 2023.
>
> >Q: Can you go into more detail on how you sampled data from TEM-8?
>
> A: We sampled data from the TEM-8 dataset randomly. We will add this information in the revision.
>
> >Q: Can you specify which metric you used to measure inter-annotator agreement?
>
> A: The inter-annotator agreement in Sec.3.2 is calculated as the proportion of samples that two annotators reach a consensus. We will make it clear in the revision.
>
> We will conduct proofreading to ensure the typo and grammar issues are fixed.  Thank you.

---

### Official Review · Reviewer_3EQP · 2023-08-04

**Soundness:** 4

**Ethical Concerns:**

Yes

**Excitement:**

4: Strong: This paper deepens the understanding of some phenomenon or lowers the barriers to an existing research direction.

**Justification For Ethical Concerns:**

- I am not sure whether the sources of TEM-8 data are authorized since the authors did not talk about them. As such, it remains uncertain if the data collection process adheres to copyright regulations and permissions.

- The paper does not provide information on how the authors handled the privacy issues associated with the test takers of TEM-8.

To address these ethical concerns, the authors should provide transparent and detailed information about the sources of TEM-8 data, and privacy consents to ensure compliance with copyright and privacy laws.

**Paper Topic And Main Contributions:**

This paper aims to examine and improve the robustness problem of grammatical error correction (GEC) systems, with a particular focus on natural context perturbations. The main contribution of this study is the collection of five perturbed variants of GEC data instances through human labor, followed by the evaluation of systems' robustness in terms of correction consistency and performance drop. In addition to annotating the conventional benchmarks (CoNLL-14, BEA-19), the authors have also created a new benchmark, TEM-8, and annotated perturbed instances. The robustness evaluation covers prevalent GEC systems (seq2seq, seq2edit, LLMs) across the three benchmarks mentioned earlier. The experimental results reveal that context perturbations can mislead the correction behaviors of current GEC systems. Furthermore, the authors propose a preliminary KL-divergence-based method to alleviate the robustness problems.

**Questions For The Authors:**

- A. For the coefficient weight $\alpha$ in Eq. 3, the authors chose not to include a balancing weight (1-$\alpha$) to the NLL loss. What are the reasons behind this choice?

- B. Line 423: “The systems collectively achieve their best context robustness on TEM-8.” I noticed that the average number of errors of TEM-8 is much lower (1.47) than other data sources (2.60, 3.12). Is it possible that a lower number of errors in the TEM-8 dataset contribute to the high robustness phenomenon of GEC systems trained on it?

**Reasons To Accept:**

- **Novelty**: This paper is novel compared to the related works in the field of GEC robustness. It is the first to explore context perturbations and introduces a novel benchmark along with an assessment protocol.
- **Implications**: The investigation into context perturbations is vital because these perturbations can be introduced by users, indicating the potential for improving the reliability of GEC systems in real-world applications.
- **Resources**: Section 3 outlines a thorough creation procedure for the new datasets, including strict quality control. The collected perturbation data and TEM-8 benchmarks are of high quality and hold valuable utility for examining robustness problems within the GEC community.
- **Methodology**: In addition to providing a robustness benchmark, this paper presents a practical solution for addressing context perturbations. Remarkably, the proposed method remains effective even for the most robust baseline (GECToR).

**Reasons To Reject:**

- **Ethics**: The paper demonstrates a thorough consideration of ethics while creating the new benchmark. However, there is a notable concern regarding copyright. TEM-8 is a language test organized by an official Chinese institution, but the authors claim that it was collected from "educational websites," without providing clarity on whether these websites are officially authorized sources. Besides, it is also unclear how the authors deal with the privacy problems of test takers of TEM-8.

- **Analysis**: Some arguments in this paper may not be clear. I list them below:
  - 1) Line 417: “Given that numerous individuals rely on LLMs as personal grammar checkers today…”. Running LLMs can indeed be expensive for individuals, and many might prefer using commercial products due to their availability and cost-effectiveness. However, the paper suggests the potential use of LLMs for GEC systems without concrete evidence of tech giants developing such systems. If the authors wish to maintain this point, they need to provide further evidence or references to support their claim.

  - 2) Line 470: “...count the number of occurrences of the target word (i.e., B) in the training data…”. According to Appendix, the reproducible GEC systems are trained/fine-tuned on the CLang8 data. But the training data of ChatGPT is unknown for now, how can we ensure the divided types of word occurrences are reasonable for ChatGPT?

**Reproducibility:**

4: Could mostly reproduce the results, but there may be some variation because of sample variance or minor variations in their interpretation of the protocol or method.

**Reviewer Confidence:**

4: Quite sure. I tried to check the important points carefully. It's unlikely, though conceivable, that I missed something that should affect my ratings.

**Typos Grammar Style And Presentation Improvements:**

- Caption: “Table 3: Results of data quality inspection” -> “Table 3: Results of data quality inspection.”
- Line 196: “1k GEC samples” -> “1,000 GEC samples”
- Line 494: “Kullback-Leibler (KL) divergence.” -> “KL divergence.” (the abbreviation already appeared in the previous section)

---

> ### Author Rebuttal · Authors · 2023-08-29
>
> ### Dear Reviewer 3EQP
> We are very grateful for your in-depth and insightful comments, which will greatly help us improve our paper. For your concerns, we provide the following responses.
> > Q: Copyright about TEM-8.
>
> A: We have consulted with professional legal advisors before collecting this data. Our advisors have confirmed that gathering exam questions from official Chinese institutional organizations on external educational websites does not infringe upon copyright laws, provided that no additional analytical content from these external sites is used. Besides, we do not use the outputs from test-takers, only gold answers are used. So there is no privacy problem for test-takers. We will make these clear in the next version. Thanks for this kind reminder.
>
> > Q: The paper suggests the potential use of LLMs for GEC systems without concrete evidence of tech giants developing such systems.
>
> A: Actually, some tech companies, such as Grammarly[1] and Google[2], have built writing assistant systems based on LLMs. Such systems can handle GEC and can be used by individuals. Moreover, individuals can query ChatGPT for GEC, which is affordable and effective, as shown in [3]. We will add such information in the revision.
>
> *Reference:*
>
> [1] Raheja, Vipul, Dhruv Kumar, Ryan Koo, and Dongyeop Kang. "CoEdIT: Text Editing by Task-Specific Instruction Tuning." arXiv preprint arXiv:2305.09857 (2023).
>
> [2] Shu, Lei, Liangchen Luo, Jayakumar Hoskere, Yun Zhu, Canoee Liu, Simon Tong, Jindong Chen, and Lei Meng. "RewriteLM: An Instruction-Tuned Large Language Model for Text Rewriting." arXiv preprint arXiv:2305.15685 (2023).
>
> [3] Fang, Tao, Shu Yang, Kaixin Lan, Derek F. Wong, Jinpeng Hu, Lidia S. Chao, and Yue Zhang. "Is chatgpt a highly fluent grammatical error correction system? a comprehensive evaluation." arXiv preprint arXiv:2304.01746 (2023).
>
> > Q: The training data of ChatGPT is unknown for now, how can we ensure the divided types of word occurrences are reasonable for ChatGPT?
>
> A: Yes. The analysis on perturbing word frequency is somewhat unreasonable for ChatGPT, as its training data is unknown. The experimental results in Figure 3(c) show that ChatGPT exhibits stable robustness for different types of word occurrences. We have mentioned this phenomenon on Line 473. We will make it clear in the revision.
>
> > Q: For the coefficient weight $\alpha$ in Eq. 3, the authors chose not to include a balancing weight (1-$\alpha$) to the NLL loss. What are the reasons behind this choice?
>
> A: Please kindly note that these two weighting methods are essentially equivalent. For $L = \alpha L1 + (1-\alpha) L2$, we can convert it to $L/(1-\alpha) = (\alpha/(1-\alpha)) L1 + L2$, and set $\beta=\alpha/(1-\alpha)$ and $L'=L/(1-\alpha)$. Then we can get $L' = \beta L1 + L2$. Hence, the selection of either method is valid. In our design, we opted for the simplicity of adding just one weight for the KL loss.
>
> > Q: Is it possible that a lower number of errors in the TEM-8 dataset contribute to the high robustness phenomenon of GEC systems trained on it?
>
> A: Yes. We also agree this could be an important factor, and will discuss this in the revision.
>
> We will meticulously revise our paper according to other helpful suggestions about typos and unclear statements. We do hope our responses can address your concerns.

---

### Official Review · Reviewer_t6Xg · 2023-08-05

**Typos Grammar Style And Presentation Improvements:** None
**Soundness:** 4

**Excitement:**

4: Strong: This paper deepens the understanding of some phenomenon or lowers the barriers to an existing research direction.

**Missing References:**

None

**Paper Topic And Main Contributions:**

This paper proposes a new dataset for improving model robustness to non-error variation in written English GEC. Specifically, the authors discuss and demonstrate the fact that several leading GEC models show marked variation in proposed corrections to the same error when the context is varied in ways that should not affect the correction. As the authors point out, this is a major concern as reliability is key to engendering confidence in GEC system users; a system that gives you different corrections when sentence context is slightly changed is not trustworthy to users. In order to demonstrate this effect, the authors annotate a new dataset containing contextual perturbations to show how GEC systems often output different, and sometimes incorrect, corrections when non-error context is varied. They draw the dataset from two existing GEC benchmarks (BEA-19 and CoNLL-14) as well as a new data source, the TEM-8 test for Chinese students of English. The start with a set of 5000 GEC sentence pairs and ask 10 annotators who are proficient (though non-native) English speakers to generate 5 perturbations for each sentence. They first use this data to test 5 different well-known GEC systems, and demonstrate that these systems all, to varying degrees, have difficulty in generating consistent corrections when faced with non-error perturbations. The authors argue that this may be evidence of these models not having actually learned the grammatical patterns of English, but rather spurious correlations that allow them to perform reasonably on the various benchmarks. The authors also develop a method of fine-tuning the non-LLM models to encourage consistency in generated corrections by penalizing larger KL divergence values between sentences which should result in the same correction. They show that this approach can substantially increase model robustness to perturbation. The authors also show that this type of fine-tuning with synthetic perturbations can also increase model robustness, though to a lesser degree. However, this means that even for situations in which annotating new data isn't feasible, the authors approach could help stabilize model output. The authors say they will release their code and dataset.

**Questions For The Authors:**

My only slight concern is that the similarity in background of the annotators could result in a bias wrt the types of contextual perturbations the annotators generate. I think the authors would do well to add a few sentences about the diversity in the perturbations to show that the perturbations are not unnaturally uniform, as this could lead to learning a new set of spurious correlations rather than actually improving learning of grammatical patterns. I'm not saying this is the case, but it is something the authors need to think about.

**Reasons To Accept:**

This is a well thought out and well written paper, and I think it is a significant contribution to the GEC community. The proposed dataset, especially, stands to benefit English GEC research. The authors raise a good point about the implications of lack of robustness to context perturbations; this may well be an indication that models are learning spurious correlations, and forcing the model to learn to output similar corrections in varied contexts may ameliorate the issue. Also, the authors experiments with synthetic data are useful as they make the authors' approach more widely accessible. Overall, I think this is a good paper and should be accepted.

Also, this annotation process was clearly a significant undertaking, and this type of data resource needs to be publicized.

**Reasons To Reject:**

I don't have any specific reasons to reject. I think this paper is well thought out and argued. And the authors were quite thorough in describing their annotation procedure (which is challenging to evaluate given the fact that the annotators are allowed to make varied contextual changes, meaning there is no way to directly compare agreement between annotators). I think that they have done a good job of explaining and defending their decisions in the annotation process.

**Reproducibility:**

4: Could mostly reproduce the results, but there may be some variation because of sample variance or minor variations in their interpretation of the protocol or method.

**Reviewer Confidence:**

4: Quite sure. I tried to check the important points carefully. It's unlikely, though conceivable, that I missed something that should affect my ratings.

---

> ### Author Rebuttal · Authors · 2023-08-29
>
> ### Dear Reviewer t6Xg
> We sincerely thank you for your great effort spent on our paper. We are also very glad to see your recognition of our work. Here are our replies to your concerns.
>
> > Q: The similarity in background of the annotators could result in a bias wrt the types of contextual perturbations the annotators generate.
>
> A: Yes, we agree. The potential bias arising from the annotator background is worth further exploration. We will add the following discussions in the revision of our paper.
>
> (1) The **demographic features** (e.g., gender, age, education) are uniformly distributed among our annotators.
>
> (2) We also analyze the **diversity** of the annotated perturbations from the linguistic aspects.
>
> |   Action   | Substitution | Insertion | Deletion |
> |:----------:|:------------:|:---------:|:--------:|
> | **Proportion** |     50.34    |   23.71   |   25.95  |
>
> In terms of the *type of actions*, we can see that despite most perturbations being substitutions, the insertions and deletions still account for a consideration portion.
>
> |   Pos-Tag   | Adjective | Adverb | Noun | Preposition | Verb | Others |
> |:----------:|:------------:|:---------:|:--------:|:------------:|:---------:|:--------:|
> | **Proportion** |   19.43    |   8.90   |   34.18  |  4.22   |  27.05  |  6.22 |
>
> As shown above, the *perturbing part-of-speech (pos) tag* is also naturally distributed.
>
> These observations could demonstrate that the perturbations in RoubstGEC are naturally uniform.
>
> (3) To enhance the diversity of our annotators' backgrounds, we additionally employ **native English speakers** to conduct annotations following your suggestions.
>
> Based on our preliminary annotation (100 cases), we compare the perturbations annotated by native and non-native speakers, and find that there is no significant linguistic differences (e.g., in edit action and pos-tag) across two group annotators.
>
> The CRS and P-CRS for BART are 41.7/83.5 on non-native-annotated data and 40.8/82.8 on native-annotated data. The difference is also marginal.
>
> These suggest that despite our annotators being non-native English speakers, their proficiency in English is high, enabling them to perform annotations similar to native speakers.
>
> Hope our responses can address your concerns.

---

### Meta-Review · Area_Chair_hiMm · 2023-09-18

**Recommendation:** 4

**Metareview:**

This paper addresses the issue of robustness in GEC – they study how the contextual changes that are not relevant to the error being corrected affect model performance, and show that state-of-the-art models exhibit variation in proposed corrections when the context is varied in ways that should not affect the correction. The paper presents a new dataset, which is a sample of 2 English GEC benchmarks, combined with data from the TEM-8 dataset. They annotated the new dataset with contextual perturbations. The paper also proposes a method of fine-tuning GEC models to avoid variation and to encourage consistency.

The paper presents solid work, and addresses an interesting topic that should be relevant to GEC researchers and those working on model robustness in NLP in general.

 Ethical concerns identified during the regular  reviewing and pointed out in the ethical reviews should be addressed.

---

### Decision · Program_Chairs · 2023-10-07

**Decision:**

Accept-Main

**Comment:**

This paper addresses the issue of robustness in GEC – they study how the contextual changes that are not relevant to the error being corrected affect model performance, and show that state-of-the-art models exhibit variation in proposed corrections when the context is varied in ways that should not affect the correction. The paper presents a new dataset, which is a sample of 2 English GEC benchmarks, combined with data from the TEM-8 dataset. They annotated the new dataset with contextual perturbations. The paper also proposes a method of fine-tuning GEC models to avoid variation and to encourage consistency.

The paper presents solid work, and addresses an interesting topic that should be relevant to GEC researchers and those working on model robustness in NLP in general.

 Ethical concerns identified during the regular  reviewing and pointed out in the ethical reviews should be addressed.